# Complex Diagnostics of Non-Specific Intellectual Developmental Disorder

**DOI:** 10.3390/ijms23147764

**Published:** 2022-07-14

**Authors:** Olga Levchenko, Elena Dadali, Ludmila Bessonova, Nina Demina, Galina Rudenskaya, Galina Matyushchenko, Tatiana Markova, Inga Anisimova, Natalia Semenova, Olga Shchagina, Oxana Ryzhkova, Rena Zinchenko, Varvara Galkina, Victoria Voinova, Sabina Nagieva, Alexander Lavrov

**Affiliations:** 1Research Centre for Medical Genetics, 115522 Moscow, Russia; genclinic@yandex.ru (E.D.); bessonovala@yandex.ru (L.B.); demina@med-gen.ru (N.D.); rudenskaya@med-gen.ru (G.R.); matyushchenko@med-gen.ru (G.M.); markova@med-gen.ru (T.M.); anisimova-inga@med-gen.ru (I.A.); semenova@med-gen.ru (N.S.); schagina@dnalab.ru (O.S.); ryzhkova@dnalab.ru (O.R.); renazinchenko@mail.ru (R.Z.); vgalka06@rambler.ru (V.G.); nagieva.sabinka@gmail.com (S.N.); alexandervlavrov@gmail.com (A.L.); 2Veltischev Research and Clinical Institute for Pediatrics of the Pirogov Russian National Research Medical University, 125412 Moscow, Russia; vivoinova@yandex.ru; 3Mental Health Research Center, 115522 Moscow, Russia

**Keywords:** intellectual disability, IDD, WES, CMA, diagnostics, exome sequencing

## Abstract

Intellectual development disorder (IDD) is characterized by a general deficit in intellectual and adaptive functioning. In recent years, there has been a growing interest in studying the genetic structure of IDD. Of particular difficulty are patients with non-specific IDD, for whom it is impossible to establish a clinical diagnosis without complex genetic diagnostics. We examined 198 patients with non-specific IDD from 171 families using whole-exome sequencing and chromosome microarray analysis. Hereditary forms of IDD account for at least 35.7% of non-specific IDD, of which 26.9% are monogenic forms. Variants in the genes associated with the BAF (SWI/SNF) complex were the most frequently identified. We were unable to identify phenotypic features that would allow differential diagnosis of monogenic and microstructural chromosomal rearrangements in non-specific IDD at the stage of clinical examination, but due to its higher efficiency, exome sequencing should be the diagnostic method of the highest priority study after the standard examination of patients with NIDD in Russia.

## 1. Introduction

Intellectual developmental disorder (IDD) is one of the most common causes of disability. According to estimates by various authors, IDD prevalence ranges from 1% to 3% [1,2] in different populations. ID disorders were defined as a group of multifactorial diseases manifesting during development and characterized by a significant decrease in intellectual functions and impaired adaptive behavior. Clinically, IDD is classified according to its severity, but the nosological classification is still being improved [3]. It has been shown that 25 to 50% of IDD cases are the result of genetic abnormalities at the chromosomal or gene level [4,5]. Genetically determined IDD is sometimes subdivided into syndromic and non-specific (NIDD). NIDD is an intellectual disability combined with non-specific phenotypic features and diffuse neurological symptoms that do not allow to establish a clinical diagnosis. Analysis of the OMIM and HGMD databases and literature data helped to identify 1711 genes and mutations, which led to monogenic diseases with IDD as a symptom. The list of genes is constantly growing due to the continual description of new forms of IDD. Interestingly, pathogenic variants in the same gene can lead to syndromic IDD and NIDD as allelic diseases [6].

Given together the genetic heterogeneity and clinical polymorphism of NIDD together, identification of the etiological factor causes significant difficulties. With the advent of high-throughput technologies, such as next-generation sequencing (NGS) and chromosomal microarray analysis (CMA), in clinical practice, it has become possible to effectively diagnose individual genetic variants of syndromic and non-syndromic IDD, as well as to identify new genes responsible for their occurrence. The development of methods of molecular genetic analysis has led to a discussion about the appropriateness of using certain genetic tests as the first-line diagnostic methods in patients with intellectual disabilities [7]. In clinical practice in Russia and some other countries, karyotype analysis with differentiated chromosome staining is traditionally used as the first-line test, while some doctors suggest using CMA or NGS [8]. Carrying out this kind of research will optimize the process of clarifying the diagnosis and will increase the effectiveness of medical and genetic counseling.

## 2. Results

### 2.1. Patients

We examined 198 affected individuals with NIDD from 171 families. A total of 47% of patients came from the Moscow area, the rest from all over Russia. A total of 14 out of 171 observations (8.2%) were familial, and the rest were isolated. The ratio of boys and girls was 1.75:1 (126 and 72, respectively). The age of the patients ranged from 6 months to 65 years.

### 2.2. Range of Variants Identified by ES

As a result of NGS, we found 17 pathogenic (Pat), 16 probably pathogenic (PP) variants, and 114 variants of unknown clinical significance (VUS) (Appendix A). Furthermore, we found two deletions during coverage analysis. No variants were found in 57 samples. After checking the segregation of variants by Sanger sequencing, de novo status and information on cis/trans phasing of the variants allowed to apply additional ACMG criteria [9] and reevaluate part of the VUS: 3 (2.6%) of them were reclassified as pathogenic, 10 (8.8%)—as likely pathogenic, 47 (41.2%) as benign, and 28 (24.5%) remained VUS. For 26 (22.8%) variants, segregation was not assessed due to the unavailability of DNA from one or both parents (Figure 1).

Thus, assessment of the segregation of the identified variant in the family increases the diagnostic efficiency of exome sequencing by 1.4 times from 19% (33 of 171) to 27% (46 of 171).

Among the identified variants, missense substitutions comprised the largest share (56%), while the variants with the formation of a premature stop codon, with a shift in the reading frame, or with a change in the conservative splicing site were less common (20, 22 and 2%, respectively). More than 90% of the identified variants arose de novo, which is consistent with the literature data on exome sequencing during NIDD [10]. Table 1 and Table 2 provide the final lists of pathogenic and likely pathogenic variants.

The diagnosis of D1020 was confirmed by MLPA.

### 2.3. Candidate Variants

Among identified VUS, three were of particular interest (Table 3). Due to segregation analysis, we could not confirm the pathogenicity of the variants. Depending on the accumulation of segregation variants, the PP1 pathogenicity criteria can be classified as supporting, moderate or strong. However, the recommendations for interpretation guidelines do not provide clear indications when a certain criterion should be supporting or strong, leaving this choice to the interpreter [9]. Despite this, we believe that these variants are likely to be the cause of the disease, but further research is needed.

### 2.4. Spectrum of Variants Identified by CMA

CMA was performed in 91 patients, 42 before NGS with a negative result, and 49 after NGS. In patient S3, heterozygous variant p.R2535X in VPS13B was identified by NGS. Since VPS13B-associated IDD is an autosomal recessive disease and the patient’s phenotype was partially overlapping with that described earlier, we decided to perform exon-level CMA to search for the second variant in this gene. There were three such patients in total, but we were able to identify the second variant in the gene only in one of them. The rest of the patients underwent high-resolution CMA. Since we did not have the opportunity to perform CMA for all patients, we selected them based on the number of minor developmental anomalies and other clinically significant symptoms described earlier [33].

We identified 22 pathogenic and likely-pathogenic CNV in 16 patients, including 13 deletions and nine duplications (Table 4). The most frequently affected chromosomes were 5, 8, and 18.

Three CNV involved only one gene (20%), and two deletions (13%) affected SHANK3, which is often described in IDD and Phelan–McDermid syndrome. Deletion on chromosome 18 in D898 was suspected during the coverage analysis and was later confirmed by CMA; the result is presented in Table 4.

### 2.5. Genotype–Phenotype Correlation

Exome sequencing identified 75% of all molecular diagnoses, 72% were autosomal dominant IDD, 17% were autosomal recessive IDD, and 11% were X-linked IDD. Chromosomal microarray analysis identified 25% of diagnoses; it is interesting that 13% of the rearrangements affected only one gene or even part of it, 47% affected only one dose-sensitive gene associated with IDD, and 40% were large rearrangements with the involvement of more than one disease-causing dose-sensitive genes.

To identify differences in the group of patients with CNV, SNV, and undiagnosed, we assessed the most common clinical and dysmorphic features found in patients. The results are shown in Table 5.

Significant differences were identified in the group of patients with microcephaly; this symptom was more common in the group of patients with SNV compared with the groups of patients with CNV and undiagnosed (*p* = 0.0016). A patent with VPS13B-associated IDD was assigned to the group diagnosed by exome sequencing.

Despite the absence of differences between the groups only in specific dysmorphisms only, their number was significantly lower (*p* = 0.00001) in the group without an established diagnosis compared with the groups of patients with established diagnoses by either CMA or NGS. Given the absence of differences in the incidence of dysmorphisms in patients diagnosed with CMA and NGS (*p* = 0.14), they were combined into one group (Figure 2).

In the groups of patients with different numbers of clinical and dysmorphic features (including dysmorphic features, epilepsy, macro/microcephaly, malformations, hypotonia, and motor delay were included in the analysis)—significant differences in the diagnostic efficiency were observed (Figure 3).

This result suggests the low efficiency of molecular genetic diagnostics in patients with isolated intellectual disabilities.

## 3. Discussion

Here we report the results of comprehensive diagnostics of 171 families with non-specific IDD families. There were no variants that were registered in more than one family. Most SNVs were not described by the time of establishing the diagnosis for the patient (25 out of 46). A total of 22 variants were identified for the first time; two of them are described in more detail in the case reports. Most CNVs were also identified for the first time. Among the diagnosed cases, autosomal dominant forms of IDD predominated; their proportion was 25% (42 out of 171), 92% of the detected mutations arose de novo, and two variants were inherited from parents with the corresponding phenotype; one variant was presumably due to maternal gonadal mosaicism. Recessive IDD comprised 5% (9 out of 171), X-linked IDD—3.5% (6 out of 171), complex chromosomal rearrangement—3.5% (6 out of 171). Although the remaining 108 cases (63% of the total number of the patients) were undiagnosed, there were several patients with VUS, which can potentially explain the clinical picture. In a similar work by Soden S.E. et al., the contribution of AR pathology was greater due to the inclusion in the sample of severe patients with hereditary metabolic diseases or lethal in the neonatal period [34]. Such patients were not included in the experimental sample of the present study. Otherwise, the distributions by type of inheritance are similar.

We identified three variants in two genes associated with Coffin–Siris syndrome (*SMARCA4, SOX4*), two variants in *ADNP* (Helsmoortel van der Aa syndrome), one variant in *SMARCA2* (Nicolaides–Baraitser syndrome), and one variant in *ACTL6B* (Intellectual developmental disorder), all of which are associated with BAF (SWI/SNF) complex [35,36]. Thus, the variants associated with the BAF complex were the most common in our patients. Notably, these patients lacked dysmorphic features or specific manifestations of any of the above-mentioned syndromes and could not be suggested for targeted gene analysis. Our findings support the idea of looking for the cause of undiagnosed non-specific IDD among other genes associated with the BAF complex but not previously associated with IDD [37].

It is interesting to note that the phenotype of these and the other seven patients with the identified variants in the genes associated with syndromic IDD (*VPS13B, RAI1, SON, POGZ, SATB2, NFIX*) upon re-examination generally fit into the classical description of the syndromes. However, the combination of phenotypic features recorded by the doctor at the time of examination did not allow to suspect the proper diagnosis prior to molecular diagnostics. Moreover, not all clinical and phenotypic manifestations of the syndrome appear simultaneously, and with age, they may worsen or weaken [38], which makes clinical differential diagnosis difficult, and in this regard, NGS is a universal diagnostic tool that allows us to confirm diagnoses in complex cases.

Most of the variants reported in this study were found in the genes described in patients demonstrating great phenotypic variability from severe syndromes with multiple clinical signs to the isolated non-specific IDDs (Appendix A). Thus all our patients with established molecular diagnoses match the corresponding syndromes or IDD types except one patient (D473 *BRD4* (NM_058243.3:c.3666_3672dup (p.Glu1225GlnfsTer16)). Patients with pathogenic variants in *BRD4* develop Cornelia de Lange-like syndrome (Appendix A). The patient from our cohort had only poor posture, narrow palpebral fissures, and small hands and feet. Mild phenotype can be explained by the distal position of the frameshift variant in the 18th exon out of 20. The total number of patients with BRD4-associated Cornelia de Lange syndrome is very small, which may limit our understanding of the phenotypic heterogeneity. We had another case with an extremely rare X-linked IDD caused by the pathogenic variants in *FRMPD4*, where the clinical features are non-specific and cannot be combined in the “typical” set or syndrome (Appendix A).

From the functional point of view we found several interesting cases including three missense variants in the genes which were previously reported to have only LoF variants in patients (D381 *TRAPPC6B* NM_177452.4:c.119G > T (p.Gly40Val); D837 *RBFOX1* NM_145891.3:c.1252T > G; (p.Tyr418Asp) and D954 *BCAP31* NM_001139441.1:c.716G > A(p.Gly239Asp)). Variants in *TRAPPC6B* and *RBFOX1* are predicted to affect splicing and consequently disrupt gene function. *BCAP31* variant seems to be a classic missense, and we failed to classify it as (likely) pathogenic, keeping VUS annotation. Moreover, there was one LoF variant (D737 TUBB NM_178014.4:c.623_624del (p.Tyr208Ter)) in the gene which is characterized by high LoF intolerance (pLI = 0.98). HGMD reports only 14 variants, of which there are 12 missense pathogenic variants and whole gene deletion and frameshift deletion with supposed deleterious effects in patients with Hirschsprung disease [39].

Almost half of the CNV included a single gene that is currently associated with IDD and is susceptible to either duplication or deletion. Such CNVs have been described for epilepsy and intellectual disability [40]. At present, these diseases cannot be classified as clearly monogenic since CNVs usually include one or more other genes that are not yet associated with phenotypes; their role in the pathogenesis of IDD is to be determined in the future. However, such a high incidence of small CNV allows us to recommend performing exon-level or high-resolution CMA in patients with NIDD.

The efficiency of NGS strongly depends on the type of diagnosed pathology; for example, osteogenesis imperfecta gives more than 50% efficiency [41,42], clinically diagnosed syndrome—32%, and autistic spectrum disorders—4% to 9% [43,44]. Similarly, for another high throughput method, CMA, it has been shown that the presence of heart defects significantly improves diagnostic efficiency [45]. In our patients with NIDD, the overall efficiency of NGS and CMA reached 35.7% (26.3% for NGS, 8.8% for CMA, and 0.6% detected by both methods). It is difficult to compare studies of non-specific IDD due to the high heterogeneity of the patient cohorts and the diversity of the inclusion criteria. For example, Taskiran et al. reported an efficiency of 49.2%, but 74.6% of patients were from consanguineous unions, which increased the detection rate [46]. In our study, patients were of different ages and ethnicities, so the data on diagnostic efficiency are more relevant for the entire population. If we compare the effectiveness of NGS in syndromic IDD, then in this case, even targeted sequencing is more efficient in this case, as almost 40% in work by Martinez et al. [47]. The lower efficiency in our study can be explained by the non-specific phenotype of patients, which can be easily confused with sporadic non-genetic IDD. The efficiency of diagnosing using CMA in non-syndromic IDD is about 11% [48], which is comparable to our indicators for the entire cohort. If we take into account only the tested patients, then the efficiency will be higher (17.5%) due to enrichment in patients with phenotypic features. Segregation analysis of the identified variants increases the diagnostic efficiency of the exome sequencing by 1.3 times from 20% to 27%. It is important to pay attention to this during genetic counseling and to suggest the family provide samples of the proband and both parents. In the case of isolated intellectual disability, full trio sequencing is the best diagnostic solution. Firstly, this is the most effective diagnostic method at the moment [49], and secondly, such an analysis will allow the parents and the physician to stop long exhausting diagnostic searches and to engage more thoroughly engage in the rehabilitation of the child. In addition, trio analysis has proven to be an ideal method for the detection of de novo variants, which constitute the majority of the cases reported previously [10] and in our study.

There were no dysmorphic features that could help the physician choose between NGS and CMA as the first diagnostic step. Only microcephaly appears to be the prognostic factor of higher efficiency of exome analysis compared to CMA (*p* = 0.0016). However, the overall efficiency of NGS efficiency is higher than that of CMA. We believe that NGS should be the standard first-line diagnostic method after the standard examination (karyotype, TMS, urinary organic acids, abnormal methylation of the *FMR1* promoter region, point mutations, and deletions/duplications in the *MECP2* gene [50]) in patients with IDD in Russia.

## 4. Materials and Methods

### 4.1. Patients

We selected 198 subjects with NIDD according to the formulated criteria from the patients of the Research Centre for Medical Genetics consulted from 2017 to 2020. Criteria included normal results of karyotyping or CMA, absence of abnormal methylation of the *FMR1* promoter region in boys, absence of data for the clinical diagnosis of any known syndrome, absence of perinatal problems, intrauterine complications, and absence of epileptic seizures before IDD manifestation. Written informed consent was obtained from healthy parents. The study was approved by the Ethics Committee of the Research Centre for Medical Genetics (protocol code 5/8 from 12 November 2018).

### 4.2. Isolation of Genomic DNA

DNA was isolated from 200 μL of whole blood using the Quick-gDNA MiniPrep Kit (ZymoResearch, Irvine, CA, USA) according to the manufacturer’s instructions.

### 4.3. NGS

NGS was carried out for 1 individual per family. We performed either whole-exome sequencing (WES) or clinical exome sequencing (CES). WES was performed by either using the Ion AmpliSeq™ Exome DRY S5 Kit 1 × 8 (Thermo Fisher Scientific, Waltham, MA, USA) according to the manufacturer’s protocol or using the TruSeq^®^ ExomeKit (Illumina, San Diego, CA, USA) according to the manufacturer’s protocol. CES was performed using the SeqCap EZ HyperCap kit (Roche, Basel, Switzerland), targeting 6500 clinically relevant genes [51]. DNA libraries were sequenced on the NextSeq 500 system (Illumina, San Diego, CA, USA).

All pathogenic, likely pathogenic variants, and variants of unknown significance were confirmed by Sanger sequencing, and segregation analysis was performed where possible.

### 4.4. Bioinformatic Analysis

The raw sequencing data were aligned to the human genome (GRCh37/hg19) using the following standard automated algorithm provided by Illumina for data analysis at https://basespace.illumina.com (accessed on 13 November 2021) or the Torrent Variant Caller algorithm provided by ThermoFisher Scientific. The identified detected variants were annotated with the ANNOVAR software. Splicing alteration prediction was carried out using the SpliceAI (https://github.com/Illumina/SpliceAI (accessed on 13 November 2021)) software, NetGene2 (https://services.healthtech.dtu.dk/service.php?NetGene2-2.42 (accessed on 13 November 2021)), and Human Splicing Finder (http://www.umd.be/HSF3/ (accessed on 13 November 2021)). Annotated variants were filtered and interpreted according to ACMG recommendations [9]. All variants in the manuscript are named according to the standard nomenclature: http://varnomen.hgvs.org/recommendations/DNA v20.05 (accessed on 13 November 2021).

### 4.5. Chromosomal Microarray Analysis

CMA was performed according to the manufacturer’s protocol using the CytoScan ™ HD and CytoScan ™ XON microarrays (Affymetrix, Waltham, MA, USA).

### 4.6. Statistical Analysis

For statistical analysis, we used the nonparametric Chi-square test with Yates’s correction, the Student’s *t*-test with Bonferroni’s correction, and the Kruskal–Wallis and Mann–Whitney tests.

## Figures and Tables

**Figure 1 ijms-23-07764-f001:**
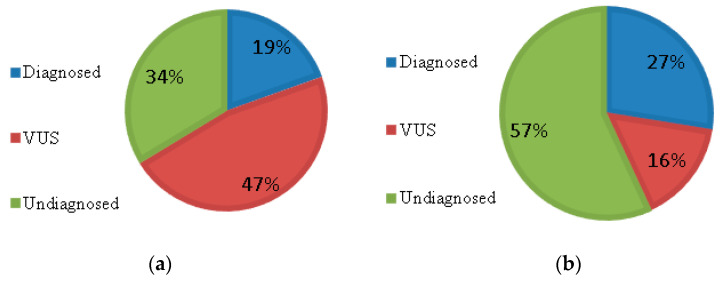
Efficiency of exome sequencing before (**a**) and after (**b**) segregation, tested by Sanger sequencing.

**Figure 2 ijms-23-07764-f002:**
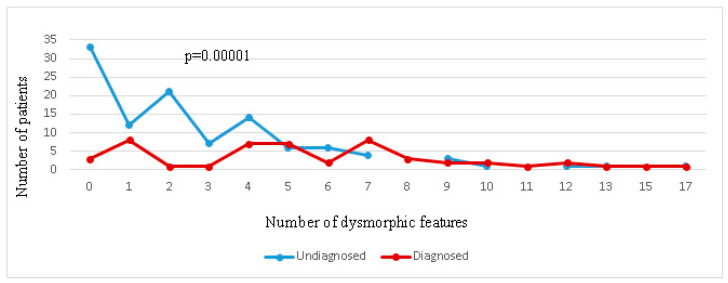
Distribution of patients with different numbers of dysmorphic features in groups with and without a molecularly confirmed diagnosis.

**Figure 3 ijms-23-07764-f003:**
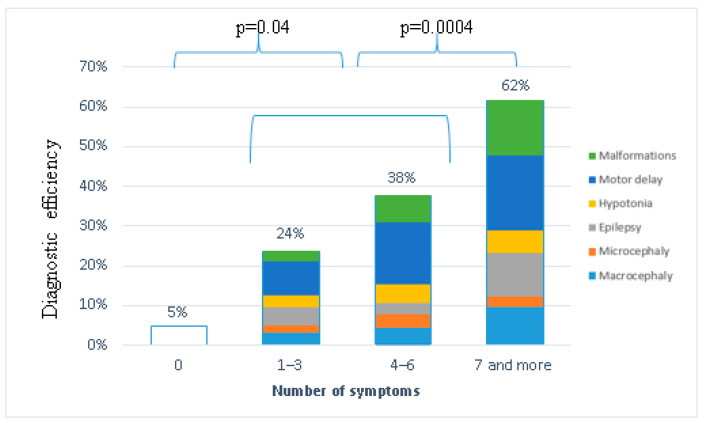
Diagnostic efficiency depending on the number of the major phenotypic features.

**Table 1 ijms-23-07764-t001:** Variants classified in HGMD as disease-causing.

	DNA id	Gene	GRCh37	Zygosity	References
1	D553	*ADNP*	NC_000020.10:g.49510461G > A NM_015339.5:c.790C > T NP_056154.1:p.(Arg264Ter)	het	[11]
2	D467	*AFF3*	NC_000002.11:g.100623270C > T NM_001025108.2:c.772G > A NP_001020279.1:p.(Ala258Thr)	Het	* [12]
3	D656	*ALG13*	NC_000023.10:g.110928268A > G NM_001099922.3:c.320A > G NP_001093392.1:p.(Asn107Ser)	het	[13]
4	D232	*GRIA1*	NC_000005.9:g.153144076G > A NM_001114183.2:c.1906G > A NP_001107655.1:p.(Ala636Thr)	het	[13]
5	D352	*HGSNAT*	NC_000008.10:g.43037306G > A NM_152419.3:c.1031G > A NP_689632.2:p.(Arg344His)	hom	[14]
6	D475	*KCNQ2*	NC_000020.10:g.62073782C > T NM_172107.4:c.793G > A NP_742105.1:p.(Ala265Thr)	het	[15]
7	D645	*MED13*	NC_000017.10:g.60108837G > A NM_005121.3:c.977C > T NP_005112.2:p.(Thr326Ile)	het	[16]
8	D835	*NDST1*	NC_000005.9:g.149921213G > A NM_001543.5:c.1831G > A NP_001534.1:p.(Gly611Ser)	hom	[17]
9	D1060	*NHLRC2*	NC_000010.10:g.115636390G > T NM_198514.4:c.442G > T NP_940916.2:p.(Asp148Tyr)	hom	[18]
10	D923	*POMGNT1*	NC_000001.10:g.46661719G > A NM_017739.4:c.385C > T NP_060209.4:p.(Arg129Trp)	hom	[19]
11	D542	*PPP2R1A*	NC_000019.9:g.52715968C > A NM_014225.6:c.533C > A NP_055040.2:p.(Thr178Asn)	het	[20]
12	D428	*PPP2R5D*	NC_000006.11:g.42975009G > A NM_006245.4:c.598G > A NP_006236.1:p.(Glu200Lys)	het	[21]
13	D189	*PTEN*	NC_000010.10:g.89717712C > T NM_000314.8:c.737C > T NP_000305.3:p.(Pro246Leu)	het	[22]
14	D341	*PTEN*	NC_000010.10:g.89692904C > T NM_000314.8:c.388C > T NP_000305.3:p.(Arg130Ter)	het	[23]
15	D957	*SMARCA4*	NC_000019.9:g.11132465C > T NM_001128849.3:c.2681C > T NP_001122321.1:p.(Thr894Met)	het	[24]
16	D293	*STXBP1*	NC_000009.11:g.130430439G > A NM_003165.6:c.875G > A NP_003156.1:p.(Arg292His)	het	[25]
17	D808	*TMEM222*	NC_000001.10:g.27660774T > C NM_032125.3:c.539 + 2T > C NP_115501.2:p.?	hom	* [26]
18	S3	*VPS13B*	NC_000008.10:g.100791008C > T NM_017890.5:c.7603C > T NP_060360.3:p.(Arg2535Ter)	het	* [27]

* The articles that describe in detail the corresponding patients listed in the table. Other articles report pathogenicity of the variants in other patients.

**Table 2 ijms-23-07764-t002:** Variants classified as pathogenic or likely pathogenic by ACMG classification.

	DNA id	Gene	GRCh37	Zygosity	ACMG Criteria	Gnomad Frequency	Structural Functional Impact #	ACMG Classification
1	D177	*ACTL6B*	NC_000007.13:g.100246360A > G NM_016188.5:c.554T > C NP_057272.1:p.(Leu185Pro)	hom	PM2, PP1(M), PP2, PP3.	n/d	LoF?	LP
2	D410	*ADNP*	NC_000020.10:g.49509097del NM_015339.5:c.2155del NP_056154.1:p.(Tyr719ThrfsTer9)	het	PVS1, PS2, PM2, PP5 [28]	n/d	LoF	P
3	D594	*AHDC1*	NC_000001.10:g.27877448_27877449del NM_001029882.3:c.1181_1182del NP_001025053.1:p. (Cys394SerfsTer122)	het	PVS1, PS2, PM2	n/d	LoF	P
4	D473	*BRD4*	NC_000019.9:g.15349980_15349986dup NM_058243.3:c.3666_3672dup NP_490597.1:p.(Glu1225GlnfsTer16)	het	PVS1, PM2	n/d	LoF	LP
5	D424	*DDX3X*	NC_000023.10:g.41198298A > G NM_001193416.3:c.113A > G NP_001180345.1:p.(Tyr38Cys)	het	PS2, PM2, PP2, PP3 * [29]	n/d	interacting with EIF4E region	LP
6	D659	*DNMT3A*	NC_000002.11:g.25468920G > T NM_175629.2:c.1443C > A NP_783328.1:p.(Tyr481Ter)	het	PVS1, PS2, PM2, PP5 (ClinVar:872726)	n/d	LoF	P
7	D375	*DYRK1A*	NC_000021.8:g.38858824_38858827del NM_001396.5:c.572_575del NP_001387.2:p.(Lys191ThrfsTer6)	het	PVS1, PS2, PM2, PP5 [30]	n/d	LoF	P
8	D843	*FRMPD4*	NC_000023.10:g.12725711G > T NM_014728.3:c.1411G > T NP_055543.2:p.(Glu471Ter)	hemi	PVS1, PS2, PM2	n/d	LoF	P
9	D289	*GRIN1*	NC_000009.11:g.140057096G > C NM_007327.4:c.1918G > C NP_015566.1:p.(Ala640Pro)	het	PS2, PM2, PP2, PP3	n/d	transmembrane helix	LP
10	D971	*HUWE1*	NC_000023.10:g.53561589G > A NM_031407.7:c.12719C > T NP_113584.3:p.(Ser4240Phe)	hemi	PS2, PM2, PP2, PP3	n/d	LoF	LP
11	D682	*MED13L*	NC_000012.11:g.116403946del NM_015335.5:c.6331del NP_056150.1:p.(Gln2111SerfsTer18)	het	PVS1, PS2, PM2 * [31]	n/L	LoF	P
12	D332	*NEXMIF*	NC_000023.10:g.73961725C > T NM_001008537.3:c.2667G > A NP_001008537.1:p.(Trp889Ter)	hemi	PVS1, PS2, PM2	n/d	LoF	P
13	D1020	*NFIX*	NC_000019.9:g.13183833_13264696del	het	PVS1, PS2, PM2	n/d	LoF	P
14	D364	*PGAP3*	NC_000017.10:g.37829376G > A NM_033419.5:c.827C > T NP_219487.3:p.(Pro276Leu)	hom	PM2, PP1, PP3, PP4, PP5 (ClinVar:426134)	6.35 × 10^−5^	transmembrane helix	LP
15	D680	*POGZ*	NC_000001.10:g.151400859dup NM_015100.4:c.600dup NP_055915.2:p.(Gly201TrpfsTer114)	het	PVS1, PS2, PM2	n/d	LoF	P
16	D198	*RAI1*	NC_000017.10:g.17696891C > G NM_030665.4:c.629C > G NP_109590.3:p.(Pro210Arg)	het	PS2, PM2, PP3.	n/d	“region” (Uniprot)	LP
17	D837	*RBFOX1*	NC_000016.9:g.7760742T > G NM_145891.3:c.1252T > G NP_665898.1:p.(Tyr418Asp)	het	PS2, PM2, PP3.	n/d	gaining of acceptor splice site	LP
18	D685	*SATB2*	NC_000002.11:g.200246475_200246476del NM_015265.4:c.414_415del NP_056080.1:p.(Val139GlyfsTer69)	het	PVS1, PS2, PM2	n/d	LoF	P
19	D886	*SATB2*	NC_000002.11:g.200233333dup NM_015265.4:c.696dup NP_056080.1:p.(Lys233Ter)	het	PVS1, PS2, PM2	n/d	LoF	P
20	D543	*SCN2A*	NC_000002.11:g.166172096_166172097del NM_021007.3:c.1499_1500del NP_066287.2:p.(Glu500AlafsTer21)	het	PVS1, PS2, PM2	n/d	LoF	P
21	D601	*SCN2A*	NC_000002.11:g.166188070G > A NM_021007.3:c.2380G > A NP_066287.2:p.(Gly794Arg)	het	PS2, PM2, PP2, PP3	n/d	repeated domain II	LP
22	D171	*SMARCA2*	NC_000009.11:g.2056756C > T NM_003070.5:c.1258C > T NP_003061.3:p.(Arg420Cys)	het	PS2, PM2, PP2, PP3	3.98 × 10^−6^	HSA domain	LP
23	D1059	*SMARCA4*	NC_000019.9:g.11134267G > A NM_001128849.3:c.2933G > A NP_001122321.1:p.(Arg978Gln)	het	PS2, PM2, PM5, PP2, PP3	n/d	no	P
24	D495	*SON*	NC_000021.8:g.34926178_34926179del NM_032195.3:c.4641_4642del NP_115571.3:p.(His1547GlnfsTer9)	het	PVS1, PS2, PM2	n/d	LoF	P
25	D965	*SOX4*	NC_000006.11:g.21595046G > A NM_003107.3:c.281G > A NP_003098.1:p.(Gly94Asp)	het	PS2, PM2, PP3	n/d	DNA-binding region	LP
26	D336	*TRIP12*	NC_000002.11:g.230657847_230657848del NM_004238.3:c.3759_3760del NP_004229.1:p.(Gly1254IlefsTer36)	het	PVS1, PS2, PM2.PP5 (ClinVar:521198)	n/d	LoF	P
27	D737	*TUBB*	NC_000006.11:g.30691462_30691463del NM_178014.4:c.623_624del NP_821133.1:p.(Tyr208Ter)	het	PVS1, PS2, PM2	n/d	LoF	P
28	D755	*ZBTB18*	NC_000001.10:g.244217659C > T NM_205768.3:c.583C > T NP_991331.1:p.(Arg195Ter)	het	PVS1, PS2, PM2, PP5 [32]	n/d	LoF	P

* The articles that describe in detail the corresponding patients listed in the table. #-Functional effects and phenotypes of patients with new SNVs are described in more details in the Appendix A.

**Table 3 ijms-23-07764-t003:** Likely causative VUS.

DNA id	Gene	Position (hg19)	Zygosity	ACMG Criteria	Gnomad Frequency	Structural Functional Impact #	ACMG Classification
S4	*DYNC1H1*	NC_000014.8:g.102446128_102446130del NM_001376.5:c.591_593del NP_001367.2:p.(Gln198del)	Het	PM2, PM4, PP1	n/d	coiled coil structural motif	VUS
D381	*TRAPPC6B*	NC_000014.8:g.39628717C > A NM_177452.4:c.119G > T NP_803235.1:p.(Gly40Val)	Hom	PM2, PP1 (moderate, PP3	n/d	cryptic donor splice site	VUS
D954	*BCAP31*	NC_000023.10:g.152966417C > T NM_001139441.1:c.716G > A NP_001132913.1:p.(Gly239Asp)	Hem	PM2, PP1 (moderate, PP3	5.74 × 10^−6^	no	VUS

# Functional effects and phenotypes of patients with new SNVs are described in more details in the Appendix A.

**Table 4 ijms-23-07764-t004:** Pathogenic and likely-pathogenic CNV.

	DNA id	Molecular Karyotype	Size Mb	IDD Genes
1	S3	46,XX,arr(hg19) 8q22.2(100286202_100287976)x3	0.002	*VPS13B*
2	D212	46,XX,arr(hg19) 1p21.2p13.3(101493397_111245231)x1	9.7	*KCNA2*
3	D254	46,XX,arr(hg19) 5q31.2q31.3(138736957_139616749)x1	0.88	*PURA*
4	D296	46,XX,arr(hg19) 17p11.2(16745570-20449778)x3	3.7	*RAI1*
5	D362	46,XX,arr(hg19) 16q23.1q24.3(78036006_90155062)x3, 18q23(77840421_78013728)x1	12.1 0.17	*ANKRD11*
6	D494	46,XY,arr(hg19) 5q14.3(88018426_88641953)x1	0.62	*MEF2C*
7	D600	arr(hg19)16q22.3-q21.3 (73919969_75197862)x1	1.2	*GLG1*
8	D801	46,XY,arr(hg19) 18p11.21 (12360754_12425280)x1	0.06	*AFG3L2*
9	D856	46,XX,arr(hg19) 10q26.3(133728056-135427143)x1 16p13.3(85881_5019217)x3	1.6; 4.9	*TSC2, CREBBP*
10	D872	46,XX,arr(hg19) 22q13.31q13.33(45676621_51177928)x1, 21q22.3(44879067_48097372)x3	5.5; 3.2	*SHANK3*
11	D898	46,XX,arr(hg19) 5p15.33p15.32(113577_5170554)x3, 18q22.3q23(70734990_78014123)x1	5 7.3	*TERT, NDUFS6* *ZNF407, CTDP1*
12	D904	46,XY,arr(hg19) 17p12(14087934_15484858)x3, 22q13.32q13.33(48571448_51197838)x1	1.4 2.6	*SHANK3*
13	D928	46,XY,arr(hg19) 2q33.1q34(202909263_211154254)x1	8.2	*MAP2*
14	D948	46,XY,arr(hg19) Xq21.1q21.31(80848988_90921090)x1	10	*ZNF711*
15	D951	46,XY,arr(hg19) 7q36.1(151934936_151936775)x3	0.002	*KMT2C*
16	D1011	46,XY,arr(hg19) 8p23.3p23.1(158049_6999220)x1 8p22p23.1(11895232_39651909)x3	6.8 27.8	*DLGAP2* *KAT6A, NEFL*

**Table 5 ijms-23-07764-t005:** Comparison of the most common clinical and dysmorphic features.

Features	Undiagnosed		Diagnosed by ES		Diagnosed by CMA	
	Amount	Fraction	Amount	Fraction	Amount	Fraction
Motor developmental delay	34	31%	23	50%	9	60%
Malformations	31	28%	13	28%	8	53%
Dysplastic ears	20	18%	12	26%	2	13%
Microcephaly (*p* = 0.0016)	11	10%	**16**	**35%**	1	7%
Seizurs	26	24%	13	28%	2	13%
Low-set ears	12	11%	8	17%	2	13%
Hypotonia	9	8%	9	20%	2	13%
Macrotia	15	14%	4	9%	0	0%
Valgus feet	6	5%	9	20%	3	20%
Number of patients	110		46		15	

## Data Availability

The data presented in this study are available upon request from the corresponding author.

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
