# Peer review of "Complex Diagnostics of Non-Specific Intellectual Developmental Disorder"

_ijms, 2022, doi:10.3390/ijms23147764_

Round 1

Reviewer 1 Report

The study describes the molecular characterization of 198 patients with non-specific IDD from 171 families using whole exome sequencing and chromosome microarray analysis. WES analysis allowed the identification of pathogenic or likely pathogenic mutations in 27% of the 171 families, while MCA in 16 of 91 tested individuals (17.5%?). Most of the identified variants are located in genes associated with the BAF (SWI/SNF) complex. Their analysis fails to find phenotypic features that allow differential diagnosis among individuals carrying point mutations and those with CNVs.

This study provides genetic data and clinical features of a great number of diagnosed individuals. 26 out of 46 SNVs and most of the CVNs were identified for the first time. However, they did not describe all the implications of their findings.

The authors should discuss if the diagnostic yield they reached is in accordance with that reported in literature, and if it is different from diagnostic yield reported in syndromic ID?

The supplementary materials contain phenotype of patients with new SNVs and comparison with previously described features. The authors should describe the results in terms of new genotype-phenotype correlation.

Among the diagnosed cases, there are some that are of particular interest?

Did you find other variants of particular interest in the diagnosed cases?

How did you menage variants in genes that have been reported both with autosomal or recessive inheritance

The authors must provide the criteria they used to classify variants as pathogenic or likely pathogenic and report in the table which variants are pathogenic or likely pathogenic.

The authors should provide possible structural functional impact of the identified variants in particular for the missense variants.

The study was approved by the Local Ethics Committee. The authors should specify which Local Ethics Committee (s).

A bioinformatics analysis of the genomic data generated by WES or CES has not been provided. Only one splicing variant has been identified. How has been conducted the analysis on possible splicing variants?

The complete list of the identified VUS variants should be provided as supplementary file.

It is very important to share genetic and phenotypic data. All the variants described in the papers should be submitted to a Public Database of genetic variants.

Author Response

We're very thankful for the comments and agree with them. We tried to answer in details the questions and to correct the manuscript accordingly. Below are our point by point answer and comments:

The study describes the molecular characterization of 198 patients with non-specific IDD from 171 families using whole exome sequencing and chromosome microarray analysis. WES analysis allowed the identification of pathogenic or likely pathogenic mutations in 27% of the 171 families, while MCA in 16 of 91 tested individuals (17.5%?). Most of the identified variants are located in genes associated with the BAF (SWI/SNF) complex. Their analysis fails to find phenotypic features that allow differential diagnosis among individuals carrying point mutations and those with CNVs.

This study provides genetic data and clinical features of a great number of diagnosed individuals. 26 out of 46 SNVs and most of the CVNs were identified for the first time. However, they did not describe all the implications of their findings.

The authors should discuss if the diagnostic yield they reached is in accordance with that reported in literature, and if it is different from diagnostic yield reported in syndromic ID?

We discussed diagnostic efficiency in more detail in the Discussion section.

The supplementary materials contain phenotype of patients with new SNVs and comparison with previously described features. The authors should describe the results in terms of new genotype-phenotype correlation.

Syndromic and non-syndromic IDD are known for the great phenotypic and genotypic variability. Many genes demonstrate both AR and AD types of inheritance, there is a lot of cases of different penetrance even inside one family. We did hope to establish some of the clinical features to help doctors or at least variant interpreters to suspect and analyze specific groups of patient in a more targeted way and we describe some of the trends allowing classification of the patients (figure 2 and 3), however there is no chance to establish any phenotype-genotype correlations, especially keeping in mind that there are no repeated variants in our patients and lots of them are de novo with zero frequencies worldwide.

Among the diagnosed cases, there are some that are of particular interest?

A couple of the most interesting cases are studied in more details and the results are published including reporting two new syndromes. Other interesting cases are described in details in the supplementary.

Did you find other variants of particular interest in the diagnosed cases?

Yes, we have accidental findings in genes of highly penetrant cancers several times. However, we did not reflect this in our work, since it does not relate to the topic of intellectual developmental disorders.

How did you menage variants in genes that have been reported both with autosomal or recessive inheritance

In these cases, we describe other evidences of different type of inheritance usually explained by the dominant negative effect with missense variants and recessive effect of the loss-of-functions evidence, for example case in D177 the variant in ACT6B would presumably lead to loss of function and a recessive form of the disease, while in D289 the variant in GRIN1A would conversely lead to gain of function and a dominant form of the disease. We have discussed this more vigorously in supplementary materials.

The authors must provide the criteria they used to classify variants as pathogenic or likely pathogenic and report in the table which variants are pathogenic or likely pathogenic.

New columns are added. Table is generally updated, description of the variants and their classification was improved in the table and in the text where appropriate.

The authors should provide possible structural functional impact of the identified variants in particular for the missense variants.

We added discussion of the functional effects in the supplementary data as well as briefly in the table.

The study was approved by the Local Ethics Committee. The authors should specify which Local Ethics Committee (s).

Corrected

A bioinformatics analysis of the genomic data generated by WES or CES has not been provided. Only one splicing variant has been identified. How has been conducted the analysis on possible splicing variants?

We added some description of the bioinformatics analysis in the Materials and Methods. We use SpliceAI, NetGene2 and Human Splicing Finder. We have identified more than one splicing variant, however, they are located in exons and require functional analysis. Also some of them in supplementary table with all VUS variants.

The complete list of the identified VUS variants should be provided as supplementary file.

We provided supplementary table with all VUS variants.

It is very important to share genetic and phenotypic data. All the variants described in the papers should be submitted to a Public Database of genetic variants.

You are absolutely right, it is very important and it was in our plans. However, this procedure is not so fast. Our organization has an institute API in ClinVar and we promise to submit all variants during July.

Reviewer 2 Report

This is an interesting article, but I have to make the following remarks:

2.1 The description of ethnic diversity in paragraph 2.1 is unacceptable in its present form. In a scientific article classifications must be specified and in relation with the goals. Ethnicity has no strong genetic basis and has above all administrative purposes. It is sufficient to write that 47% of patients came from the Moscow area and the others from all over Russia.

2.2 It is appropriate to specify the description of the different modalities by which the segregation study by Sanger sequencing made it possible to reclassify the pathogenicity of the variants.

2.3 The discussion about the pathogenicity of the four new variants discovered by the authors is well conducted. However, in table 2, I see only 3 variants.

Figure 2 – Add the title of the x-axis coordinate: number of dysmorphic features.

Figure 3 and Lines 147 – 153. The question posed is interesting but it miss some explanations about the constitution of groups. For example, multi-coloured cumulative histogram of different phenotypic features could be informative.

Line 153: 5 % sensibility is a low efficiency of molecular genetic diagnostics, extremely low is usually restricted to test-sensitivity inferior or equal to 1%.

3 – line 158 “Most SNVs were not previously described (26 out of 46)”. I do not understand this assertion, there seems to me a discrepancy with line 88 of the article : “The variants with an asterisk were described with the participation of the authors of this article”. There are four asterisks. Please provide further and clearer explanations.

Supplementary data - Previously undescribed genetic variants, a brief description of the phenotype

A beginning of explanations to the remark done on the preceding paragraph is given in the supplementary data. However, these phenotypic data seem to me to be insufficient to take the heavy decision of including a new variant in a table of pathogenic or likely pathogenic variants. Please also provide a more complete argument on genotypic data and their correlations with phenotypes.

Author Response

We're very thankful for the comments and agree with them. We tried to answer in details the questions and to correct the manuscript accordingly. Below are our point by point answers and comments:

2.1 The description of ethnic diversity in paragraph 2.1 is unacceptable in its present form. In a scientific article classifications must be specified and in relation with the goals. Ethnicity has no strong genetic basis and has above all administrative purposes. It is sufficient to write that 47% of patients came from the Moscow area and the others from all over Russia.

Corrected

2.2 It is appropriate to specify the description of the different modalities by which the segregation study by Sanger sequencing made it possible to reclassify the pathogenicity of the variants.

We have added ACMG pathogenicity criteria to the variant table.

Trio analysis allows to establish de novo status (PS2) cis/trans (PM3, BP2) or sometimes analysis of several family members allows to establish co-segregation with the disease (PP1). Explanation is added in the text.

2.3 The discussion about the pathogenicity of the four new variants discovered by the authors is well conducted. However, in table 2, I see only 3 variants.

Sorry, we have corrected this error and replaced 4 with 3 in the text.

Figure 2 – Add the title of the x-axis coordinate: number of dysmorphic features.

Corrected

Figure 3 and Lines 147 – 153. The question posed is interesting but it miss some explanations about the constitution of groups. For example, multi-coloured cumulative histogram of different phenotypic features could be informative.

We added multicolor histogram.

Line 153: 5 % sensibility is a low efficiency of molecular genetic diagnostics, extremely low is usually restricted to test-sensitivity inferior or equal to 1%.

Thank you, we fixed that mistake.

3 – line 158 “Most SNVs were not previously described (26 out of 46)”. I do not understand this assertion, there seems to me a discrepancy with line 88 of the article : “The variants with an asterisk were described with the participation of the authors of this article”. There are four asterisks. Please provide further and clearer explanations.

This manuscript summarises the results of the 4-years study, part of which were presented elsewhere in details describing new phenotypes. However, during the interpretation process these variants were found to be associated with IDD for the first time. For the consistency of the manuscript we divided the table 1 in two parts – previously clearly described (likely) pathogenic variants and variants were described in ID for the first time with the notes, that some of them were already described by us in details elsewhere.

Corrected table.

Line 168-169: Most SNV were not described by the time of establishing the diagnosis for the patient (25 out of 46)

Supplementary data - Previously undescribed genetic variants, a brief description of the phenotype

A beginning of explanations to the remark done on the preceding paragraph is given in the supplementary data. However, these phenotypic data seem to me to be insufficient to take the heavy decision of including a new variant in a table of pathogenic or likely pathogenic variants. Please also provide a more complete argument on genotypic data and their correlations with phenotypes.

We added to each case a variant HGVS name and a patient number for easy correlation with the data in Tables 1 and 2, which we also reformated to provide a clearer classification of pathogenecity of the variants including formal ACMG criteria.

Round 2

Reviewer 1 Report

I think the autors improved the English of the manuscript in the revised version and that they addressed some of the points I highlighted in my first report. However, I have further consideration and I am still not satisfied about the presentation and discussion of the main findings of this work.

I still believe that the supplementary materials contain the main interesting findings of this work and that the main text did not give the right space to them.

The authors should describe in which genes your findings confirm the association of the gene with a broader phenotypic spectrum, and in which cases the broader spectrum has not been highlighted in literature. Also, are there cases in your cohort that much very well with the final diagnosis?

The authors should describe in which dosage-sensitive genes, where loss of function variants are known to cause the disease, they found a pathogenic missense variant.

Here some minor points to be addressed:

Revised the Bioinformatics analysis in the Methods section.

In the “Supplementary Materials” Section:

Patient D332. NEXMIF NM_001008537.3:c.2667G>A; NP_001008537.1:p.(Trp889Ter)

The variant p.(Trp889Ter) disrupts the neurite extension and migration factor that is involved in neurite outgrowth by regulating intercellular adhesion via the N-cadherin signaling pathway.

The authors should add a reference.

Patient D336. TRIP12 NM_004238.3:c.3759_3760del;

without epiactivity and seizures, had a frame-shift variant p.Arg1253fs., leading to the…

The word “epiactivity” should be corrected.

Patient D473. BRD4 NC_000019.9:g.15349980_15349986dup; NP_490597.1:p.(Glu1225GlnfsTer16)

In the literature, 5 patients with pathogenic variants in the BRD4 have been described [23,24]

Patient D843. FRMPD4 NM_014728.3:c.1411G>T; NP_055543.2:p.(Glu471Ter)

The 104-th type of X-linked IDD is one of the rare ones; only 10 patients from 4 unrelated families have been described.

Since there are few patients described in literature, this finding should be mentioned in the main text. I do not agree using the pictures of the patients published elsewhere to describe similarities in dysmorphic features. The readers can go to the original publication to confirm this statement. The authors should describe in the text the dysmorphic similarities.

in a girl with atypical Ret syndrome [40] described as NM_006352:c.C556T:p.(R186X) in ZNF238.

Please correct with Rett Syndrome….

Patient D965. HUWE1 NM_031407.7:c.12719C>T; NP_113584.3:p.(Ser4240Phe) Variants in the HUWE1 gene cause an X-linked dominant disease, the clinical features of which are moderate to profound NIR,

Please explain the acronym “NIR”

Exome sequencing identified a de novo missense substitution p.Tyr418Asp. The SpliseAI [43] predictor program predicted a possible gain of the splicing acceptor site that can lead to possible loss of function of RNA binding protein fox-1 homolog 1.

Please correct SpliceAI

Reviewer 2 Report

The authors have responded to my original comments and the manuscript has improved considerably. The article is a compilation of 4 years research on this topic and is worth reporting. But there are so many changes in this new version that the article becomes messy. Please, further improve the simplicity and clarity of the article as a whole and the discussion in particular

Author Response

The authors have responded to my original comments and the manuscript has improved considerably. The article is a compilation of 4 years research on this topic and is worth reporting. But there are so many changes in this new version that the article becomes messy. Please, further improve the simplicity and clarity of the article as a whole and the discussion in particular

Please check the new version with accepted changes and corrected minor mistakes